# The Burnout Syndrome among Women Working in the Retail Network in Associations with Psychosocial Work Environment Factors

**DOI:** 10.3390/ijerph18115603

**Published:** 2021-05-24

**Authors:** Gintarė Kalinienė, Dalia Lukšienė, Rūta Ustinavičienė, Lina Škėmienė, Vidmantas Januškevičius

**Affiliations:** 1Department of Environmental and Occupational Medicine, Public Health Faculty, Lithuanian University of Health Sciences, LT-47181 Kaunas, Lithuania; dalia.luksiene@lsmuni.lt (D.L.); ruta.ustinaviciene@lsmuni.lt (R.U.); lina.skemiene@lsmuni.lt (L.Š.); vidmantas.januskevicius@lsmuni.lt (V.J.); 2Health Research Institute, Faculty of Public Health, Lithuanian University of Health Sciences, LT-47181 Kaunas, Lithuania; 3Laboratory of Population Studies of the Institute of Cardiology, Medical Academy, Lithuanian University of Health Sciences, LT-47181 Kaunas, Lithuania

**Keywords:** burnout, psychosocial work environment, retail workers

## Abstract

The burnout syndrome is a significant occupational health problem in various employees’ populations. The aim of this study was to evaluate burnout level among retail network workers and its associations with psychosocial work environment. The cross-sectional epidemiological study was conducted on workers of one Lithuanian retail network (*n* = 254), where all respondents were women. In order to assess their occupational stress and burnout, two instruments were used: HSE management standards work-related stress indicator tool and Copenhagen burnout inventory (CBI). The statistical analysis showed high prevalence of burnout—the frequency of personal, work-related and client-related burnout was 53.5%, 66.5% and 55.5% respectively. The Spearman’s correlation analysis revealed that job demands, control manager’s support, coworkers’ support and relationships significantly associated with all burnout subscales. The multivariable logistic regression analysis was performed to determine the independent associations between HSE indicators and burnout subscales. The multivariate logistic regression model revealed that job demands and manager’s support were significant factors for all burnout dimensions. In conclusion, in order to reduce occupational burnout among employees working in retail companies, it would be useful for occupational interventions to focus on workload reduction and optimization, and for the human resources management strategy to focus on maintaining this.

## 1. Introduction

Work-related stress has been widely investigated in recent decades by scientists because it is one of the most frequently reported work-related health problems in Europe [1,2]. Sickness absence attributed to work-related stress and the number of people suffering from stress-related conditions caused or made worse by work are likely to increase [2,3]. The burnout syndrome may be defined as the intermediate state between negative work environment factors and disease in the modern occupational stress understanding, where work stress is defined as the process by which workplace psychosocial stressors produce both primary (strain, tension and anxiety) and secondary (mental or physical diseases) effects [4]. However, burnout is a syndrome that results from chronic stress at work and usually is described as a combination of symptoms including energy depletion or exhaustion, increased mental distance from one’s job, or feelings of negativism or cynicism related to one’s job, and reduced professional efficacy [5]. This is in accordance with the historical development of the burnout concept, where burnout is described as a state of physical, emotional and mental exhaustion that results from long-term involvement in work situations that are emotionally demanding [6]. Despite the widespread use of Maslach burnout conception in the scientific field of burnout evaluation, which assesses burnout syndrome through evaluating three dimensions (emotional exhaustion, depersonalization and reduced professional efficacy [7]), the instrument is criticized as not reflecting the real concept of burnout, as not applicable to any occupational context, and as not being cross-cultural. All three dimensions mentioned above should be measured independently rather than utilizing a simpler, unidimensional score [6,8,9]. For all these reasons, in order to assess the prevalence of work-related burnout among trade workers as alternative for MBI scales, the Copenhagen burnout inventory (CBI) was chosen. In the CBI, the core of burnout is fatigue and exhaustion—the aspects that we aim to reveal in our study.

The burnout problem, and work-related stress, is investigated insufficiently in the retail sector. Health, education or public services employees are more commonly investigated in this regard in comparison with other occupations. There is no doubt that employees of these occupational groups are working in the most adverse environment in regard to psychosocial work conditions, because the prevalence of burnout and other related conditions is very high [10,11]. In spite of that, burnout more or less affects employees from all occupational settings and industries [2,3,12,13]. A recent Polish cross-sectional study conducted among the employees of commercial service sector and teachers unexpectedly showed that burnout is at the same level among the employees of both groups [14]. The investigation conducted in a sporting goods store also showed a high level of burnout among staff: this strongly correlated with occupational stress and negatively correlated with social support [15]. Moreover, the study of employees working in a large supermarket chain showed high prevalence of low job support, low job control and high job demands [16].

Sellers must maintain strong bonds with their consumers as a necessity of business. Workers in service jobs constantly interact with customers and deal with their arguing, complaints and sometimes excessive demands [17]. Emotional labor is a job stressor that leads to burnout and increased stress levels [18]. It was confirmed that salespeople’s regulation of emotions is conducive to reducing interpersonal conflict and felt stress, which eventually leads to higher performance [19]. However, employees of the commercial service sector show a greater tendency toward choosing surface acting as regulation of emotions strategy. Unfortunately, surface acting leads to the increase in burnout [14]. In addition, the emotional dissonance as a discrepancy between required and felt emotions is a characteristic phenomenon for people whose duties are directly associated with consumer service and was found to be an occupational predictor for adverse health effects [20].

Work family conflict (or work–life conflict) is another important source of occupational stress among various retail companies the main causes of which are high job demand, long and unpredictable working hours, short and split shifts and the need to work multiple jobs to earn a living [21]. Work family conflict consequences may be negative to an organization because of deviant behavior of workers towards the retailer, coworkers, and/or customers [22].

We want to highlight the importance of occupational stress investigations among various occupational groups of employees. Therefore, in the present study we chose the seller consultants of a clothing retail company because, as it was mentioned before, scientific studies confirmed many specific occupational features that causes occupational stress and burnout. Our hypothesis is that work in direct contact with costumers, work according to the imposed model of behavior and high workload may cause adverse effects, such as occupational burnout. The aim of this study was to evaluate burnout level among women working in a retail network and the associations of this burnout with psychosocial work environment.

## 2. Materials and Methods

### 2.1. Procedure and Participants

The cross-sectional epidemiological study was conducted on workers of one Lithuanian retail network in May 2018 to assess their occupational stress and burnout. The study was approved by the Bioethics center of LUHS and approval to perform the study was provided (Nr. BEC—vs. (M)—115). 

The participants were selected by following one of the instruments (HSE management standards indicator tool) used in the study manual (guide) [23]. According to HSE instrument manual, 50% of the investigated population should be included in the survey to ensure proper representativity. As the number of working seller consultants at the retail network in Lithuania was 504, 252 participants are a sufficient sample size. A random sample of 345 seller consultants working with customers was composed and a digital version of anonymous questionnaires was disseminated through the retail organization’s network. In all, 259 respondents answered the questionnaire. Only five males answered the questionnaire, so due to a small number, we excluded males from the analysis. Finally, 254 women were included into the analysis (response rate was 75%). Other characteristics of respondents are presented in Table 1.

### 2.2. Instruments

Two standardized questionnaires were used in order to compose the final questionnaire for the survey. The first questionnaire was the HSE management standards work-related stress indicator tool [24], approved and validated version for the Lithuanian population [25]. It is composed of 35 items and covers seven domains of psychosocial work environment: demands (e.g., I have unachievable deadlines; I have to work very intensively), control (e.g., I can decide when to take a break; I have a choice in deciding how I do my work), manager’s support (I am supported through emotionally demanding work; I can talk to my line manager about something that has upset or annoyed me about work), colleagues’ support (e.g., If work gets difficult, my colleagues will help me; I receive the respect at work I deserve from my colleagues), relationship (e.g., I am subject to bullying at work; relationships at work are strained), role (clarity) (e.g., I am clear what is expected of me at work; I am clear what my duties and responsibilities are) and changes (how organizational change is managed and communicated) (e.g., staff are always consulted about change at work; when changes are made at work, I am clear how they will work out in practice). Each scale is made up of a certain combination of questions, and each answer uses five-point Likert’s scale (from “Never” to “Always” or from” “Strongly Disagree” to “Strongly Agree”). Answers were transformed into scores ranging from 0 (poor) to 100 (desirable). An overall scale score was computed as the sum score across questions in each scale and was divided into three groups based on the margins of terciles, categorized as 1st tercile named low/weak/poor; 2nd tercile—average; 3rd tercile—high/strong/clear. Cronbach’s alpha reliability for all HSE (management standards work-related stress indicator tool) scales was found to be between 0.65 and 0.84: the scale was considered as reliable when the Cronbach’s alpha was bigger than 0.6 [26].

The second questionnaire used in the survey was intended to measure the burnout of the retail network seller consultants. For this purpose, we chose the standardized questionnaire of Kristensen TS with coauthors—Copenhagen burnout inventory [8]. This instrument includes three domains of burnout: personal burnout (5 items, e.g., How often do you feel tired? How often do you think: “I can’t take it anymore”?), work-related burnout (10 items, e.g., Is your work emotionally exhausting? Does your work frustrate you?) and client-related burnout (6 items, e.g., Does it drain your energy to work with clients? Are you tired of working with clients?). Answers were also transformed into scores ranging from 0 to 100. Total score on the burnout scale is the average of the scores on the items. Personal burnout, work-related burnout, and client-related burnout in the analysis were defined as dichotomized variables with a cut-off point at the 50 scores (scores higher than 50 points indicate the presence of burnout). The calculated Cronbach’s alphas for all three scales also showed good reliability—0.77–0.89.

Questions that covered demographic data (age, gender, work experience and workload) were also included in the final questionnaire.

### 2.3. Statistical Analysis

The data was analyzed with the Statistical Package for Social Science (SPSS) 20.0 (IBM Inc., Armonk, New York, NY, USA). Respondent demographics were reported using descriptive statistics. The Kolmogorov–Smirnov test was used to verify the normality of the variables. The Spearman’s correlation was computed to estimate the direct or indirect association between the variables with statistical significance level *p* < 0.05. For the evaluation of the associations between psychosocial work environment factors and dependent variables—personal, work and client-related burnout, a binary logistic regression analysis was applied. Firstly, univariate logistic regression analysis was performed: just one independent was included in the model (Model 1). An odds ratio (OR) was calculated with a 95% confidence interval and *p* value and presented in Model 1 (Table 4). Secondly, a binary multivariable logistic regression analysis was performed to determine the independent associations between HSE indicators and burnout subscales (Model 2). In this model, all HSE indicators (demands, manager’s support, colleagues’ support, relationship, role (clarity) and changes) and personal characteristics (respondents’ age, work experience and workload) as independent variables were included in multivariable logistics regression analysis. Additionally, an odds ratio (OR) was calculated with a 95% confidence interval and *p* value and presented in Model 2 (Table 4).

## 3. Results

All 254 respondents included in the survey were women. The mean age of the respondents was 26.91 ± 6.02 years (Table 2). The mean duration of work experience in this retail network was 4.76 ± 4.46 years. More than a half (57.1%) of respondents were full day workers in this job (Table 1). Descriptive statistics for each of the HSE indicator tool variable and the CBI subscales’ personal, work-related and client-related burnout values are shown in Table 1, which includes means, standard deviations, median, minimum and maximum values.

The calculated dichotomized variables of three burnout dimensions and general burnout variable (representing all three) scales showed high prevalence of burnout. More than half of respondents (53.5%) were classified as a high personal burnout perceived group, 66.5% as a high work-related burnout perceived group and 48.3% as a high client-related burnout perceived group. In general (including all three scales) 55.5% of respondents were classified as experiencing perceived burnout.

The correlations between outcome variables (personal, work-related and client-related burnout) and independent variables HSE dimensions, also demographic characteristics are shown in Table 3. Respondents’ age was found to be significantly related to personal burnout, and the workload was related to personal and client-related burnout. Demands, control manager’s support, coworkers’ support and relationships significantly associated with all burnout subscales.

The odds ratios of personal burnout, work-related burnout and client-related burnout according to the psychosocial work environment factors are presented in Table 4. Firstly, univariate logistic regression analysis (Model 1) was performed. The results showed that all burnout subscales were significantly associated with work demands. The ORs of burnout increase with average and high levels of work demands in comparison with a low level: the sellers with the average and highest level of work demand had significantly higher probability of personal (OR = 3.03 and OR = 7.51 respectively), work-related burnout (OR = 3.07 and OR = 6.31 respectively) and client-related burnout (OR = 3.92 and OR = 7.36 respectively). The managers’ support dimension also displayed significant associations with all burnout subscales (Model 1). The workers with strong managers’ support had a significantly lower probability of personal, work-related and client-related burnout (OR = 0.19, OR = 0.19 and OR = 0.18 respectively). Workers with the average managers’ support also had a significantly lower probability of work-related burnout (OR = 0.41). The reduced probability of suffering from work-related burnout significantly associated with coworkers’ support: the average and strong coworkers’ support reduced work-related burnout by about 30% (OR = 0.33 and OR = 0.36 respectively) (Model 1). Moreover, sellers with average and poor relationships at work had a greater probability of suffering personal burnout (OR = 2.19 and OR = 4.36) and work-related burnout (OR = 2.81 and OR = 2.65). The changes in the workplace significantly associated with work-related burnout: the respondents who perceived workplace changes as high had a 52% lower work-related burnout probability (Model 1).

Next, in order to evaluate the effect of HSE indicators for burnout subscales, all predictors and respondents’ age, work experience and workload were entered in the multivariate regression model (Model 2). These results revealed that the work demands dimension remained significant in the multivariate regression analysis and displayed the strongest associations with burnout subscales in comparison with other HSE indicator tool dimensions. Additionally, strong managers’ support remained significant for all burnout subscales. Only average coworkers’ support remained significant for work-related burnout. However, HSE dimensions, such as relationships and changes, did not show significant associations with CBI subscales.

## 4. Discussion

In recent years, burnout research has been focused on women and young persons [27,28,29,30]. Studies show that women experience more burnout than men, and the factors that cause occupational stress and burnout for women and men are different [31,32]. At the same time, the degree of burnout decreases with increasing age and length of service. Studies showed that burnout negatively related with income, positively with university education and position in the organization [33]. All respondents in our study were women, because traditionally mostly women work in the field of clothing sales. We have a situation where women are the main workforce at sales companies. Creating a favorable working environment without stress is essential for success in sales. By evaluating factors that influence stress and burnout for each group of workers, we can create good psychosocial work environment for sale workers. At the same time, it would allow businesses to achieve a customer-friendly environment.

There are a lot of studies of burnout in nurses, physicians, managers and teachers populations, but a limited number of investigations of people working in sales companies, in direct contact with clients. So, the aim of our study was to evaluate the employees of this often underappreciated work. This type of work is not considered difficult and complicated by public opinion. This type of work does not involve physical, ergonomic or chemical factors in the work environment. However, we found unexpectedly high levels of stress and burnout, which indicates a particularly unfavorable psychosocial environment. Our results show that 55.5% of respondents were classified as perceived burnout, and according to our data, it was related mainly with work environment. These findings are very similar to the results from other studies, which evaluate professions with greater responsibilities, higher education level and more complex types of work, such as healthcare workers or teachers [31,34].

Each profession has a different work environment, a different level of stress and burnout and different factors of the work environment that determine burnout. A person working as a salesperson experiences emotional strain, degrading customer behavior and lack of managerial support [27]. According to publications, the most important factors for sales workers were job insecurity, lack of social support, inappropriate behavior in the workplace and job demand [27,35,36].

In our study, the investigated population was of a young age and with relatively short work experience in the sales company. Mean age of the investigated population was 26.91 years and work experience was 4.76 years. The study of working women in Shanghai provides the results that young-aged working women had a higher occupational burnout; however, they complained less about health than older ones [29]. These research results suggest that in order to improve working conditions and quality of life for working women, the stress coping strategy for each age group must be different. Younger women may lack experience and social support to cope with occupational stress, which often leads to job dissatisfaction [30]. We can see from our study that the main factor for burnout is the psychosocial environment, namely high work demands. It is strongly related to customer service, mandatory satisfaction of their needs and avoidance of conflict situations. This is confirmed by the data of our study: high work demands play the leading role and especially closely associate with client-related burnout. Working with clients leads to depersonalization, necessity of certain emotions and the need to work as required by the service manual. So, it leads to emotional dissonance, and after some time to burnout, as it is suggested by studies conducted on persons working in the service sector [20,37,38]. Burnout and stress at work can later lead to depressive symptoms [35]. It is well-known that job insecurity due to changing non-standard working conditions has a harmful effect on women’s future physical and mental health [27]. In regard to high prevalence of burnout syndrome in various working populations [39] and its proven effect on workers’ health [40], burnout was included in the 11th Revision of the International Classification of Diseases (ICD-11) as an occupational phenomenon, but still it is not classified as a medical condition [7]. 

In human resource management, which is a prerequisite for a company’s success, an important part of human resource management is the selection, training and evaluation of individuals with the competencies necessary to perform the job. Moreover, organizations should consider the psychological resistance and emotional compatibility of staff when selecting employees and forming teams.

Our study shows that strong support at work from managers showed a positive association with all types of burnout and average and poor relationships at work related with a higher probability of burnout symptoms. Experience with interventions directed at psychosocial organizational factors shows that structural factors, such as work needs, support and control, are particularly important for burnout prevention [32,41]. It is confirmed by studies in different countries and organizations [41,42,43]. The results of our study suggest strategies for organizational changes in companies. Creation of supportive and collective work environment in the companies can be an effective strategy for preventing burnout symptoms among sales workers. Therefore, we can conclude that assessment of the psychosocial environment and its essential elements is a key direction in creating a healthy and safe workplace.

Strengths: This is the first and the only study conducted among sales network workers in Lithuania. The study represents psychosocial work environment in a whole one trade network sales company that includes all Lithuania regions. This research adds knowledge in assessing stress and burnout reactions in different occupations and their causes. It also explores the psychosocial work environment of a widespread and multi-employee group that is not, in principle, widely explored.

Limitations: The main limitation of this study is that the results should be considered only as associations rather than as causal inferences because the data on HSE indicator tool and burnout were obtained cross-sectionally. It would be better to perform a follow-up study in order to reveal a causal relationship between psychosocial work environment and burnout. The other limitation is that the respondents of the study were only women, so the future investigations with the male population are needed. It is important to state that we analyzed work conditions in the largest clothing retail network in Lithuania that has stores in the largest Lithuanian cities and district centers. It would be interesting to evaluate the situation in other retail companies in the future.

## 5. Conclusions

The average and highest level of work demand had a significantly higher probability of personal, work-related burnout and client-related burnout for women working in the sales company. However, strong managers’ support reduced the probability of burnout. According to our results, future efforts should be made to support a healthy psychosocial work environment as an occupational health priority in service and retail sectors.

## Figures and Tables

**Table 1 ijerph-18-05603-t001:** Characteristics of respondents.

		*n*	%
**Age groups**	≤25	139	54.7
	>25	115	45.3
**Daily working time duration**	Half a time	109	42.9
	Full time	145	57.1
**Work experience, years**	≤3	135	53.1
	>3	119	46.9
**Marital status**	Married	58	22.8
	Divorced	12	4.7
	Single	64	25.3
	Partnership	120	47.2

**Table 2 ijerph-18-05603-t002:** Descriptive statistics of individual factors and psychosocial stressors and burnout subscales.

	Cronbach’s Alfa	Mean	SD	Median	Min/Max
***Individual factors***					
Age	-	26.91	6.02	25	19/49
Work experience	-	4.76	4.46	3	1/31
***HSE (psychosocial stressors)***					
Demands	0.76	54.66	9.81	56.25	25/81.25
Control	0.65	48.68	14.69	50	8.33/91.67
Manager’s support	0.82	64.23	16.4	65	25/100
Co-workers’ support	0.82	67.44	16.7	68.75	18.75/100
Relationships	0.79	18.55	16.29	12.5	0/93.75
Role	0.84	81.85	13.89	85	25/100
Change	0.72	63.52	16.58	66.67	16.67/100
***CBI (Burnout subscales)***					
Personal burnout	0.89	46.83	19.81	50	0/91.67
Work-related burnout	0.77	53.01	16.27	53.57	0/89.29
Client-related burnout	0.81	42.37	17.44	47.21	0/79.35

**Table 3 ijerph-18-05603-t003:** Correlations between respondents’ age, work experience, workload, HSE dimensions and CBI subscales.

	Personal Burnout	*p*	Work-Related Burnout	*p*	Client-Related Burnout	*p*
Age	0.141 *	0.024	0.039	0.531	0.124	0.691
Work experience	0.092	0.144	−0.021	0.740	0.032	0.254
Workload	0.128 *	0.042	−0.016	0.803	0.153 *	0.038
Demands	0.449 ***	<0.001	0.362 ***	<0.001	0.451 ***	<0.001
Control	−0.173 **	0.006	−0.236 ***	<0.001	−0.321 ***	<0.001
Manager’s support	−0.345 ***	<0.001	−0.425 ***	<0.001	−0.418 ***	<0.001
Co-workers’ support	−0.194 **	0.002	−0.274 ***	<0.001	0.285 ***	<0.001
Relationships	0.396 ***	<0.001	0.343 ***	<0.001	0.414 ***	<0.001
Role	0.026	0.681	−0.014	0.823	−0.113	0.523
Changes	−0.100	0.113	−0.212 ***	0.001	−0.176	0.215

* *p* < 0.05; ** *p* < 0.01; *** *p* < 0.001.

**Table 4 ijerph-18-05603-t004:** The odds ratios of personal burnout, work-related burnout and client-related burnout according to psychosocial work environment factors.

	Personal Burnout						Work-Related Burnout							Client- Related Burnout				
**Variables**		**Model 1**			**Model 2**			**Model 1**			**Model 2**			**Model 1**			**Model2**	
	**OR**	**95%CI**	***p***	**OR**	**95%CI**	***p***	**OR**	**95%CI**	***p***	**OR**	**95%CI**	***p***	**OR**	**95%CI**	***p***	**OR**	**95%CI**	***p***
**Age**	**1.84**	**1.11–3.04**	**0.018**	0.79	0.33–1.93	0.609	1.60	0.94–2.72	0.084	1.39	0.54–3.57	0.491	1.23	0.89–2.33	0.073	1.19	0.61–2.57	0.398
**Work experience**	**1.70**	**1.03–2.80**	**0.037**	0.97	0.41–2.30	0.941	0.99	0.59–1.67	0.962	0.44	0.17–1.14	0.091	0.79	0.65–1.98	0.962	0.36	0.12–2.14	0.291
**Workload**	**1.72**	**1.04–2.84**	**0.034**	1.55	0.77–3.13	0.219	0.97	0.57–1.64	0.898	0.97	0.46–2.05	0.934	0.87	0.45–2.64	0.758	0.95	0.51–2.44	0.785
**Work demand**																		
Low	1			1			1			1			1			1		
Average	**3.03**	**1.54–5.96**	**0.001**	**2.96**	**1.38–6.34**	**0.005**	**3.07**	**1.60–5.91**	**0.001**	**3.87**	**1.78–8.41**	**0.001**	**3.92**	**1.50–6.11**	**0.001**	**3.12**	**1.69–9.14**	**0.001**
High	**7.51**	**3.70–15.3**	**<0.001**	**8.05**	**3.41–19.0**	**<0.001**	**6.31**	**3.09–12.9**	**<0.001**	**7.66**	**3.14–18.7**	**<0.001**	**7.36**	**3.56–11.2**	**<0.001**	**7.02**	**3.61–16.7**	**<0.001**
**Work control**																		
Low	1			1			1			1			1			1		
Average	1.09	0.58–2.07	0.786	1.36	0.62–2.97	0.442	0.98	0.49–1.95	0.943	1.19	0.52–2.74	0.683	0.89	0.39–1.83	0.857	1.02	0.69–3.04	0.538
High	0.68	0.38–1.20	0.182	0.61	0.29–1.29	0.198	0.59	0.32–1.09	0.095	0.62	0.28–1.36	0.230	0.54	0.22–2.09	0.092	0.59	0.36–1.99	0.330
**Manager’s support**																		
Weak	1			1			1			1			1			1		
Average	0.53	0.28–1.00	0.050	**0.41**	**0.18–0.94**	**0.034**	0.94	0.45–1.98	0.866	0.94	0.38–2.31	0.884	0.81	0.33–2.82	0.688	0.87	0.35–1.91	0.848
Strong	**0.19**	**0.10–0.36**	**<0.001**	**0.10**	**0.04–0.28**	**<0.001**	**0.19**	**0.10–0.37**	**<0.001**	**0.20**	**0.07–0.57**	**0.002**	**0.18**	**0.09–0.45**	**<0.001**	**0.19**	**0.10–0.96**	**0.001**
**Co-workers’ support**																		
Weak	1			1			1			1			1			1		
Average	0.70	0.34–1.42	0.321	1.03	0.42–2.54	0.949	**0.33**	**0.16–0.71**	**0.005**	**0.35**	**0.13–0.92**	**0.033**	0.60	0.36–1.71	0.133	0.35	0.13–1.12	0.303
Strong	0.60	0.35–1.04	0.070	1.76	0.73–4.23	0.210	**0.36**	**0.20–0.65**	**0.001**	0.85	0.33–2.21	0.735	0.56	0.20–1.65	0.362	0.52	0.23–1.91	0.573
**Relationships**																		
Good	1			1			1			1			1			1		
Average	**2.19**	**1.19–4.04**	**0.012**	0.86	0.41–1.82	0.699	**2.81**	**1.50–5.29**	**0.001**	1.20	0.54–2.65	0.660	1.71	0.91–5.29	0.069	1.38	0.71–3.35	0.606
Poor	**4.36**	**2.23–8.52**	**<0.001**	2.19	0.95–5.02	0.065	**2.65**	**1.36–5.16**	**0.004**	1.06	0.45–2.51	0.900	2.33	0.81–5.16	0.098	1.09	0.52–2.87	0.897
**Role**																		
Not clear	1			1			1			1			1			1		
Average	0.95	0.53–1.70	0.867	0.59	0.28–1.24	0.163	1.29	0.69–2.41	0.419	1.17	0.53–2.56	0.704	1.45	0.89–3.23	0.697	1.35	0.14–1.96	0.704
Clear	1.15	0.62–2.14	0.651	0.80	0.36–1.78	0.582	0.95	0.50–1.80	0.869	0.89	0.38–2.07	0.777	0.83	0.60–1.63	0.589	0.91	0.28–1.97	0.856
**Changes**																		
Low	1			1			1			1			1			1		
Average	1.08	0.55–2.13	0.824	1.69	0.85–2.41	0.716	1.21	0.57–2.59	0.619	2.76	1.00–7.62	0.051	1.51	0.41–3.61	0.519	1.76	0.31–5.26	0.125
High	0.74	0.42–1.30	0.298	1.04	0.12–1.20	0.228	**0.52**	**0.29–0.93**	**0.028**	1.55	0.60–4.01	0.364	0.63	0.15–1.46	0.093	1.63	0.59–5.09	0.428

## Data Availability

The data presented in this study are available on request from the corresponding author.

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
