# Peer review of "The Burnout Syndrome among Women Working in the Retail Network in Associations with Psychosocial Work Environment Factors"

_ijerph, 2021, doi:10.3390/ijerph18115603_

Round 1
Reviewer 1 Report
Thanks a lot for providing me with the opportunity of reviewing this work submitted to your journal for potential consideration.
After carefully reading and assessing it, I have the following observations:
1) English is very poor and should be revised.
2) It is surprising that in a paper on burnout syndrome no references are present on the classical work and definition by Maslach.
3) The theoretical framework employed is not clear and should be more explained and detailed.
4) More details should be provided about burnout syndrome epidemiology among trade workers.
5) Could authors explain why they have selected this particular population?
6) Some details about the Copenhagen Burnout Inventory should be provided.
7) A table with major socio-demographic variables should be provided.
8) Discussion should be enriched.
Author Response
Dear Reviewer,
We would like to thank you for your very valuable comments and new experiences regarding our manuscript ‘The burnout syndrome among women working in retail net-work in associations with psychosocial work environment factors’ (Manuscript ID: ijerph-1205647). We have revised the manuscript according to your comments We hope that the quality of the revised version of our manuscript has been improved as suggested. Also, we would like to inform you that an English language of manuscript was deeply revised by the professor of English language. Because there were many corrections, we didn’t highlight them for better understanding of other corrections, which are highlighted in "yellow".
I sincerely thank you on behalf of all of the co-authors.

Reviewer 2 Report
The number of participants in the study is low to evidently conclude such a result. Moreover, as mostly women participated in the study, it is very inconclusive. It is essential to specify how many of 254 participants were men or were all participants women?
Overall the study reflects a pattern between socioeconomic status and length of service including age in a sales job, however, it will be interesting to know the same for men in a similar sector if not the same.
As the study reflects women's burnout more prominently, shouldn't the title be altered accordingly? It is misleading if all the participants were women.
Author Response

(The authors gave the same response as above.)

Reviewer 3 Report
The authors have studied burnout syndrome and its association with psychosocial factors in a sample of workers of a trade network in Lithuania. This article is interesting and adds to the current literature on burnout syndrome. My comments and suggestions follow.
- What is “X trade network”? Please clarify this in the introduction section.
- “employs of these occupational groups” please correct “employs” to “employees”
- Please include your hypotheses in the introduction section. What is the gap of knowledge that this study aims to address? (please clarify this in the introduction section)
- What was your inclusion/exclusion criteria?
- It would be helpful if you provide a sample item of each questionnaire in the method section.
- “The Spearman’s correlation was computed to estimate the direct or indirect association between the variables.” What was your significance level? Did you adjust it for multiple comparisons? Please clarify this in the statistical analysis section.
- “All respondents included in the survey were women because only few males answered the questionnaire and they were not included in the analysis.” This should be included in the limitation section of the manuscript as a limitation. Also, please clearly state in the abstract that all the participants were females.
- “study was conducted on workers of one Lithuanian trade network “X”” This a major limitation for this study that affects its generalizability. Please add this to the limitation section.
Author Response
Dear Reviewer,
We would like to thank you for your very valuable comments and new experiences regarding our manuscript ‘The burnout syndrome among women working in retail network in associations with psychosocial work environment factors’ (Manuscript ID: ijerph-1205647). We have revised the manuscript according to your comments We hope that the quality of the revised version of our manuscript has been improved as suggested. Also, we would like to inform you that an English language of manuscript was deeply revised by the professor of English language. Because there were many corrections, we didn’t highlight them for better understanding of other corrections, which are highlighted in "yellow".
I sincerely thank you on behalf of all of the co-authors.

Reviewer 4 Report
This cross-sectional study analyzed burnout levels among 254 trade network workers in 2018 and their associations with the psychosocial work environment. Workers’ burnout is an important issue. However, the rationale and interpretations of their findings are lacking. In addition, the writing may not be appropriate in several parts.
(1) Rationale is lacking—A comprehensive literature related to the research topic is needed. A brief, more elaborated literature review can help understand how the studied occupational characteristics were treated in the existing studies on the association with each type of burnout. This helps clarify the extent of the novelty of the present work. In addition, the authors didn’t provide a clear rationale about why they selected this trade company or service industry. A literature review on the relationship between occupational characteristics and burnout among workers in the service industry may be more important, focusing on what is known and what is not. I feel a more elaborated literature review could help understand the importance of the present study. The authors should review the related occupational risks for trade workers and the service industry. The authors clarify why this research topic is worthy of study and how this research can contribute to the body of already existing research.
(2) Interpretations of their findings is lacking—The discussion section addressed something not directly relevant to their finding. The authors should give some interpretations and discuss the meaning of their findings. For example, the authors mentioned “young” workers, but the findings should age was positively associated with burnout. The explanation seems to contradict their findings.
(3) The is a small-scale, cross-sectional, and observational study. A more modest conclusion is recommended.
(4) Extensive editing of English language is required. For example, in the abstract, lines 16-20, there should be several punctuation marks to separate elements and to clarify the meaning of the sentence. In the second example in lines 87-88, I wonder whether the authors wanted to say “tertile” rather than “tercile.” In the third example is in line 99, the authors used “assed,” which is not an appropriate word. Fourth, please use “dot” as a decimal separator.
(5) Line 20: There is an extra %.
(6) Lines 31-33: Reference #1 is out-of-dated to support the statement in this sentence.
(7) Lines 39-42: Please provide appropriate citations to support the statement in this sentence.
(8) Lines 47-48: Please provide appropriate citations to support the statement in this sentence.
(9) Line 57: What does “X trade network” mean? If “X” is to avoid showing the company name, a description of the characteristics of this company should be provided.
(10) Line 80: Citation information of reference #14 should be translated into English.
(11) Line 90: English should be checked.
(12) Line 102: “was” should be “were”?
(13) Line 105: “statistical package for Social Science (SPSS)” should be “Statistical Package for the Social Sciences (SPSS)”.
(14) Lines 117-119: Please provide the numbers of pre-exclusion, exclusion, and after-exclusion.
(15) Line 120: I could not find “4.79 ±4.47 years” in Table 1. Instead, Table 1 indicates “4.76 ±4.46 years.” Inconsistent?
(16) Lines 120-121: Please provide numbers related to the sentence “More than a half (57.1 %) of respondents were full day workers in this job” in Table 1.
(17) Lines 175-178: It’s unclear why the authors put these two sentences here.
(18) Lines 193-194: Please provide appropriate citations to support the statement in this sentence.
(19) Lines 209-211: The authors stated there were “several studies” but only cited one reference (#19).
(20) The representativeness of studied subjects should be discussed.
Author Response

(The authors gave the same response as above.)

Reviewer 5 Report
Dear authors,
I need you to answer some questions:
INTRODUCTION
- The introduction is very short. The constructs and concepts necessary to understand the manuscript are not explained.
- The authors do not explain why the target population may suffer from burnout.
MATERIALS AND METHODS
Procedure and Participants:
- What was the target population? How was the sample chosen? The authors must specify it.
- What type of workers made up the sample? It is important to know which workers are subject to psychosocial stress.
REFERENCES
- Many bibliographies are obsolete. The bibliographic citations used are more than 5 years old 40 %). The authors must update and arrange the bibliography.
- Some references do not meet the journal guidelines.
- Some references are incomplete or have errors. The authors should review this section.
Author Response

(The authors gave the same response as above.)

Round 2
Reviewer 1 Report
Authors have extensively revised the manuscript, which can now be accepted.
Author Response
Dear Reviewer,
We would like to thank you for the positive decision. The English language of the manuscript was revised again by the language professional, who's is native English.
I sincerely thank you on behalf of all of the co-authors.
Reviewer 3 Report
The authors have addressed the comments/suggestions raised during the first round of review.
Please indicate in the statistical analysis section whether the p values are corrected for multiple comparisons in Model 1 and 2.
Author Response
Dear Reviewer 3,
We would like to thank you for the remark regarding the statistical analysis of our manuscript “The burnout syndrome among women working in the retail net-work in associations with psychosocial work environment factors” (Manuscript ID: ijerph-1205647).
Q: Please indicate in the statistical analysis section whether the p values are corrected for multiple comparisons in Model 1 and 2.
A: In response to your question about the correction of p values, our independent variables were selected purposefully based on the scientific literature results in this analysis. No search for random factors was performed. For this reason, no multiple comparisons between Models 1 and 2 were performed. We have supplemented the statistical analysis section with logistic regression analysis explanations (lines 187-197).
“For the evaluation of the associations between psychosocial work environment factors and dependent variables – personal, work, and client-related burnout, a Binary logistic regression analysis was applied. Firstly, univariate logistic regression analysis was performed: just one independent was included in the model (Model 1). An odds ratio (OR) was calculated with a 95% confidence interval as well as p value and presented in Model 1 (Table 4). Secondly, a Binary multivariable logistic regression analysis was performed to determine the independent associations between HSE indicators and burnout subscales (Model 2). In this model, all HSE indicators (demands, manager’s support, colleagues’ support, relationship, role (clarity), changes), and personal characteristics (respondents’ age, work experience, and workload) as independent variables were included in multivariable logistics regression analysis. Also, an odds ratio (OR) was calculated with a 95% confidence interval as well as p value and presented in Model 2 (Table 4).”
The English language of the manuscript was revised again by the language professional whose native language is English.
Reviewer 4 Report
Thanks for the authors' responses and intensive revision.
Author Response

(The authors gave the same response as above.)

Reviewer 5 Report
Dear authors,
Thanks for your reply. The explanations of the authors are satisfactory. Congratulations on your work.
Best regards
Author Response
Dear Reviewer,
We would like to thank you for the positive decision.
I sincerely thank you on behalf of all of the co-authors.